# The Duration of the Trial Influences the Effects of Mineral Deficiency and the Effective Phytase Dose in Broilers’ Diets

**DOI:** 10.3390/ani12111418

**Published:** 2022-05-31

**Authors:** Mehran Javadi, Alba Cerisuelo, María Cambra-López, Judit Macías-Vidal, Andrés Donadeu, Javier Dupuy, Laura Carpintero, Pablo Ferrer, Juan José Pascual

**Affiliations:** 1Institute for Animal Science and Technology, Universitat Politècnica de València, Camino de Vera s/n, 46022 Valencia, Spain; m.javadi2012@yahoo.com (M.J.); macamlo@upvnet.upv.es (M.C.-L.); jupascu@dca.upv.es (J.J.P.); 2Centro de Investigación y Tecnología Animal, Instituto Valenciano de Investigaciones Agrarias, 12400 Segorbe, Spain; ferrer_pabrie@gva.es; 3Departamento de I+D+i, Global Feed S.L.U., Grupo Tervalis, Av. Francisco Montenegro s/n, 21001 Huelva, Spain; judit.macias@tervalis.com (J.M.-V.); andres.donadeu@tervalis.com (A.D.); javier.dupuy@tervalis.com (J.D.); laura.carpintero@tervalis.com (L.C.)

**Keywords:** broilers, phytase, digestibility, mineralization, trial duration

## Abstract

**Simple Summary:**

The aim of this work was to investigate the effects of reducing the mineral (Ca and P) content and the addition of different doses of a new 3-bacterial phytase in broiler diets in a short- and long-term experiment. Mineral deficiency reduced growth performance only in young animals, and it increased Ca and P digestibility and reduced tibia mineralization, especially in the long-term trial. The effective dose of the new phytase varied with the response criteria and duration of the trial. Phytase increased feed efficiency at 500 FTU/kg, but only in young animals, and increased mineral digestibility and retention at a lower dose (250 FTU/kg) in the short- compared with the long-term (500 FTU/kg) trial. Tibia mineralization increased with only 250 FTU/kg in the long-term trial. Therefore, the age of the animals and duration of the trial are key in determining the effects of mineral levels and phytase addition in broiler feed and should be taken into account for future trials.

**Abstract:**

Two trials varying in duration (short- and long-term) were conducted to evaluate the effects of providing deficient (NC) or sufficient (PC) Ca and P levels, and different doses of a new phytase (250, 500, and 1000 FTU/kg feed), in broiler feed on growth performance, nutrient digestibility and retention, and tibia mineralization. A total of 80 and 490 male chicks (Ross) of 21 and 1 days of age were used in the short- and long-term trials, respectively. In the long-term trial, chicks fed NC diets showed a lower (*p* < 0.05) average daily gain and feed intake compared to chicks fed PC and a greater (*p* < 0.05) feed conversion ratio compared to 500 and 1000 FTU/kg feed during the starting period. Regarding the effects on minerals’ and nutrients’ coefficients of retention, animals fed NC showed a significantly higher digestibility for P than those fed the PC diet in the long-term trial. Additionally, feeding 250 to 500 FTU/kg diets increased most of the nutrients’ digestibility in the short-term but only P digestibility in the long-term trial. Tibia mineralization increased linearly with phytase addition (*p* < 0.05) only in the long-term trial. In conclusion, the effects of dietary mineral and phytase levels on growth performance are more noticeable in young animals. In addition, the duration of the trial is key due to a possible adaptation phenomenon of birds to low P supplementary levels.

## 1. Introduction

Among the minerals necessary in poultry, phosphorus (P) and calcium (Ca) are the most important, not only because they are required for an optimal growth rate, but also for bone mineralization. Phosphorus participates in metabolic processes and nutrient absorption, besides being one of the most expensive minerals in the feed’s final cost [1]. A deficiency in Ca and P, or inadequate level of Ca to P ratio, affects bone growth and development. Specifically, reduced P in broiler diets has detrimental effects on growth performance and bone mineralization [2,3,4]. Phytic acid (PA) contributes 60% to 90% of the total P in cereals, oil seeds, and nuts [5] and is considered the largest reservoir of this element in plants. Due to the lack of endogenous enzymes to hydrolyze phytate, P present in plants is biologically unavailable for monogastrics. As a result, it is necessary to add inorganic P to poultry diets to meet P requirements, increasing the cost and environmental impact of these diets, or to include enzymes, such as phytases.

Phytases are enzymes extensively used to improve the availability of P. Many studies [6,7,8,9,10,11] have shown that microbial phytases in P-deficient diets release the phytate-bound P and improve the utilization of P and other nutrients in plant-derived ingredients. Consequently, an increase in weight gain and bone ash percentages in broilers is observed. The effective dose of phytase for broilers in the literature ranges from 250 to 12,000 FTU/kg, with the inclusion level around 500 FTU/kg being the most common [7,8]. The effective dose considered as the minimum at which positive effects are observed is generally adjusted according to different response parameters, such as growth performance, mineral digestibility, bone mineralization, health, and also, recently, inositol liberation in the digesta and its concentration in plasma [9,10,12,13,14,15]. However, the interaction of phytases with these parameters is variable among studies. For example, Batabunde et al. [13] found a quadratic effect of phytase on BW, feed intake, tibia ash, and a linear response in the apparent ileal digestibility of energy and nutrients. Other studies [16,17] also observed that birds’ performance reached a plateau when the phytase level was around 500 to 1000 FTU/kg in the diet. In contrast, Shirley and Edwards [18] reported maximum performance and P retention in birds consuming up to 12,000 FTU/kg in the diet. 

Some of the factors that affect the animals’ response to phytase are dietary Ca and phytate content, experiment length, bird age, and phytase dose [9,19,20,21]. In terms of experiment length, some experiments in laying hens demonstrated that prolonging the duration of phytase supplementation using P-deficient diets had negative effects on P retention. In this regard, Bougouin et al. [9] stated that for each extra 10 d over the 92 d mean duration of layer experiments, phytase-induced P retention decreased by 0.47 percentage units. Although this effect has been generally associated with age, previous studies suggested that compensatory effects can appear, reducing the efficacy of phytases when Ca- and P-deficient diets are administrated for long periods of time to layers [22]. Studies in broilers also suggest that feeding low-P diets during long feeding periods might reduce phytase efficacy in terms of P digestibility due to homeostatic adaptations in the digestive ability of the birds that increase P retention [19,23,24]. However, other response criteria, such as growth performance or tibia mineral retention, are improved when low-P diets with phytases are fed during long periods [24].

In this context, the general objective of this study was to evaluate the consequences of reducing Ca and P content and adding different doses of a new three-bacterial phytase in broiler P-deficient diets on growth performance, nutrient, and mineral utilization and retention, and to study the possible compensatory effects and optimum phytase doses by comparing a short- with a long-term experiment. 

## 2. Materials and Methods

### 2.1. Animals and Housing

Two trials varying in duration were performed consecutively, a short-term and a long-term trial. In the short-term trial, 80 male chicks (Ross) of 21 days of age with a mean body weight (BW) of 872 ± 27.4 g (standard deviation; SD) were used. The study lasted 17 days in total. At 21 days of age, chicks were randomly allocated to 10 floor pens (1.3 × 1.3 m^2^; 8 animals/pen) located in an environmentally controlled room, and individually identified with a wing tag. Pens contained wood shavings to a depth of 10 cm and were provided with a single feed trough. A nipple watering line was provided for each of 6 pens, with 3 nipples per pen. On day 7 of the study (28 days of age), the 80 birds were moved in pairs (similar weight) to 40 metabolism cages (54 × 56 cm^2^; 8 cages/treatment; 16 animals/treatment) for a period of 10 days.

In the long-term trial, 490 male chicks (Ross) of 1 day of age with a mean BW of 45.0 ± 0.93 (SD) g were used. The study lasted 42 days in total. At day 1 of age, chickens were randomly allotted to 35 floor pens (1.3 × 1.3 m^2^; 14 animals/pen), located in two environmentally controlled rooms (20 and 15 pens in room 1 and 2, respectively). Pens contained wood shavings to a depth of 10 cm and were provided with a single feed trough. A nipple watering line was provided, with 3 nipples per pen. As in the short-term trial, 80 birds in total (2 to 3 birds per pen; 16 birds per treatment) were selected at 28 days of age according to their weight (average weight within each pen) and housed in pairs (similar weight) in 40 metabolic cages (54 × 56 cm^2^; 8 cages/treatment; 16 animals/treatment) for a period of 10 days.

In both trials, cages were provided with a feed trough with two spaces and a single nipple drinker. On the last three days in the metabolic cages (35 to 38 days of age), the total amount of excreta produced per cage was daily collected for the nutrient balance and retention study. On day 38 of the study, one chick randomly selected from each cage (8 animals/treatment) was slaughtered for bone analyses. In the long-term trial, animals that were not used for the nutrient balance and retention study were maintained in pens with 11–12 animals/pen until the end of the study (42 days of age) for performance calculations, health status, and mortality records. Additionally, in the long-term trial, chicks randomly selected from cages for bone analyses were also blood-sampled.

Throughout the study, room temperature was controlled, decreasing from 30 °C on day 1 to 20 °C on day 42 of rearing. The applied light regime consisted of 0 h of darkness followed by a period of 24 h light on d 1 and 2, and a progressive increase in the time of darkness until reaching 8 h of darkness on d 14 of the study.

### 2.2. Experimental Diets

Experimental diets were formulated based on corn, wheat, and soybean meal to fulfill the requirements of a grower feed in the short-term trial and starter and grower feeds in the long-term trial. Dietary treatments were provided from day 21 of age in the short-term trial and day 1 of age in the long-term trial. These consisted of five different diets varying in the mineral level and the inclusion of phytase (FTU/kg of feed): PC, positive control without phytase and with total Ca (0.97% for starter and 0.95% on av. for grower diets) and total P (0.65% for starter and 0.63% on av. for grower diets), levels recommended by FEDNA [25] for growing broilers; NC, negative control without phytase and with low (below requirements) levels of Ca (0.73% for starter and 0.69% on av. for grower diets) and total P (0.58% for starter and 0.51% on av. for grower diets); and another three diets for which the NC diet was supplemented with ePhyt 1000^®^ 3-phytase (Global Feed) at 250 (P250), 500 (P500), and 1000 (P1000) FTU/kg feed, respectively. Details regarding the used phytase are provided in the study by Salaet et al. [26]. In brief, ePhyt 1000^®^ 3-phytase is a bacterial phytase encoded in *Serratia odorifera* and cloned in *Komagataella phaffii*. It is stable from pH 3.7 to pH 5.8. The NC-based diets (NC, P250, P500, P1000) were produced from a common ingredient mixture to ensure the minimum variation in composition among them, and divided into 4 parts. Each part was then supplemented with the corresponding phytase doses. The ingredients and chemical composition of the experimental diets are presented in Table 1. 

Feed and water were provided ad libitum during the experimental period and feed was provided in mash form. Phytase was added to feeds in liquid form in the mixer.

### 2.3. Growth Performance

In the short-term trial, chicks were weighed by group at 21 days of age and individually at 28 days of age (allocation in metabolic cages), 35 days of age (beginning of the collection period), and 38 days of age (end of the collection period). In the long-term trial, chicks were weighed by group at d 1 of age and weekly over the trial. At 28 d of age (allocation in metabolic cages), birds were individually weighed and identified with a wing tag. In both trials, feed intake was registered at each weighing control. Body weight and feed intake were used to calculate the average daily gain (ADG), average daily feed intake (ADFI), and feed conversion ratio (FCR). In the short-term trial, ADG, ADFI, and FCR were calculated during the cage phase (from 28 to 38 days of age), and in the long-term trial, they were calculated during the whole (day 1 to 42) rearing period. 

Health status of the animals was checked daily, and necropsies were performed on all dead animals. 

### 2.4. Nutrient Retention

In both experiments, a digestibility trial was carried out to determine the coefficient of retention (CR) of Ca, P, and main nutrients, as well as Ca and P retention and excretion using metabolic cages. Sixteen animals/treatment of 28 days of age were used. Animals were adapted for 7 days to metabolic cages. On the last 3 days, feed intake and total excreta output were measured quantitatively per cage for the determination of dry matter (DM), ash, organic matter (OM), crude protein (CP), gross energy (GE), Ca, and P. During the 3-day excreta collection period, excreta were collected every 24 h and weighed, as described by Dersjant-Li [27], and maintained at 4 °C. At the end of the collection period, excreta were pooled by cage and homogenized. Representative samples were then obtained and stored at −20 °C until analyses. A representative sample of the different diets was also collected before the start of the trial to analyze their composition.

### 2.5. Bone and Blood Sampling

In both experiments, at the end of the digestibility trial, one chick per cage (8 animals per treatment) was euthanized by stunning and exsanguination to obtain tibia bone. The left tibia was removed from each bird. After removing all the soft tissues, the tibia was frozen at −20 °C until analyses to determine tibia weight, dry matter (DM), ash, Ca, and P content, as described by Dersjant-Li [27]. Additionally, in the long-term trial, blood samples from each animal were collected into 4 mL vacutainer tubes with serum clot activator. Approximately 2 h after extraction, blood samples were refrigerated and transported to the laboratory. The tubes were centrifuged for 4 min at 4000× *g* and serum was collected to determine Ca and P content.

### 2.6. Analytical Methods

Feed samples were dried at 105 °C for 24 h and then ground. Excreta samples were dried at 80 °C for 48 h and then ground. DM (934.01), ash (942.05), ether extract (920.39), and CP (990.03) determinations were carried out according to AOAC (2000) [28] procedures. Gross energy was determined using an adiabatic bomb calorimeter (Gallenkakmp, London, UK). Mineral (Ca and P) content in feeds and excreta was analyzed by inductively coupled plasma atomic emission spectrometry (ICP-OES) (model Varian 720-ES, Varian Inc., Palo Alto, CA, USA), as described in Cambra-López et al. [29]. Phytate-P in PC and NC feeds was analyzed by spectrophotometry according to the method described by Haugh and Lantzch [30]. 

For the determination of minerals (ash, Ca, and P) in bone, tibias were boiled in order to remove the remaining soft tissues, cleaned, and dried at 110 °C for 12 h; afterwards, tibias were left in an ether solution for 48 h. Once cleaned and degreased, tibias were dried again at 110 °C for 12 h, weighed, and then introduced into a porcelain crucible and ashed at 550 °C for 12 h in a muffle furnace. Mineral (Ca and P) content in tibia bones was then analyzed as previously described, adding 0.05 g ashed sample to the acid solution instead of 0.1 g as in feed and excreta.

### 2.7. Statistical Analyses

The coefficients of DM, organic matter (OM), GE, CP, Ca, and P retention were calculated using the following equation [22]: $$CR\;\left(\%\right)=\frac{\left[\left(Feed\;intake\times Nutrient_{feed}\right)-\left(Excreta\;output\times Nutrient_{excreta}\right)\right]}{\left(Feed\;intake\times Nutrient_{feed}\right)}\times 100$$
where *feed intake* is the amount of feed consumed by cage in 3 days (g), *Nutrient_feed_* is the concentration of nutrients in feed, *excreta output* is the amount of excreta produced by cage in 3 days (g), and *Nutrient_excreta_* is the concentration of nutrients in excreta.

Mineral retention was calculated as the amount of minerals ingested multiplied by their *CR*. Ash and mineral content in tibias was expressed as the percentage or absolute amount of ash, Ca, and P per tibia, once degreased and dried. 

Data were analyzed using SAS System software (Version 9.1, SAS Institute Inc., Cary, NC, USA). The pen was the experimental unit for ADG, ADFI, and FCR, and the cage for nutrient balance traits. For mineral retention in tibia and blood parameters, the bird was considered the experimental unit. The statistical model was performed using the GLM procedure of SAS, and included the diet (PC, NC, P250, P500, and P1000) as the main effect and room (1 and 2) as a block factor. Additionally, polynomial orthogonal contrasts were applied to test the linear effects of phytase level. In growth performance analyses, the initial weight of the animals was used as a covariate. The percentage of dead animals among treatments was compared using a chi-square test (FREQ procedure). Statistical significance level was set at 5% (*p* < 0.05).

## 3. Results

### 3.1. Growth Performance

In general, the health status of the animals was adequate in both trials. In the short-term trial, one chick was dead upon arrival, and in the long-term trial, a total of 28 animals died during the trial. In the long-term trial, beak abnormalities were detected in one chick (treatment NC) and three animals (treatments PC, 250, and 1000) showed leg disorders at the end of the rearing period. The percentage of dead animals was not significantly different among treatments in the long-term trial (2.38, 9.52, 5.95, 9.52, and 5.95 % for PC, NC, P250, P500, and P1000, respectively; *p* = 0.442). In the short-term trial, the diet given during the 17 days of the trial had no significant effect on any of the growth traits registered (Table 2).

However, significant differences between treatments were found in the long-term trial (Table 3). In this trial, chicks fed with the PC diet showed a greater ADFI and ADG (+3.0 ± 0.7 g/d; *p* = 0.024) than those on the NC diet during the starting period, but differences disappeared during the growing period and no differences were observed during the whole experimental period. Regarding the phytase level, chicks fed with P500 and P1000 diets showed a lower ADFI (on av. −5.1 ± 0.8 g/d; *p* < 0.001) and lower FCR than those on NC and P250 diets during the starting period (linear relationship, *p* < 0.05). No differences were observed, again, among treatments with phytase and NC during the growing period. Animals on the P500 diet had a lower ADFI (−6.2 ± 1.6 g/d; *p* = 0.031) and better FCR (−0.08 ± 0.01 g/d; *p* = 0.013) than those on the NC diet during the whole period.

### 3.2. Nutrient Utilization

Table 4 shows the effect of the experimental diets on the CR of main nutrients and on the excretion and retention of Ca and P. Diet had a great effect on nutrient utilization in the short-term trial. Chicks fed with the PC diet showed greater Ca and P excretion (+0.40 ± 0.06 and +0.21 ± 0.04 g/d animal, respectively; *p* < 0.001) but a similar CR of nutrients and Ca and P retention to those on the NC diet. Regarding the phytase level, chicks fed with the NC diet showed a significantly lower CR for DM, OM, and Ca than those fed phytase-added diets, lower CR for CP and P than those on P250 and P500, and lower CR for GE than those on the P500 diet. Consequently, animals on the NC diet showed lower Ca retention than those fed with phytase-added diets (on av. −0.53 ± 0.04 g/d animal; *p* < 0.01) and lower P retention than those on P500 and P1000 diets in the short-term trial (−0.09 ± 0.03; *p* < 0.05). The CR of DM, OM, Ca, and Ca and P retention increased, linearly, with phytase addition. In the case of the long-term trial, chicks fed with the PC diet showed also greater Ca and P excretion (+0.46 ± 0.05 and 0.25 ± 0.03 g/d animal, respectively; *p* < 0.001), but also greater Ca retention (+0.12 ± 0.04 g/d animal; *p* < 0.001) and a lower CR for P than those on the NC diet. Regarding phytase levels, animals fed with the NC diet showed lower Ca retention (−0.18 ± 0.04 g/d animal; *p* < 0.001) than those on P500 and P1000 diets, and a lower CR for P and P retention (−0.07 ± 0.03 g/d animal; *p* < 0.05) than those on P500. The Ca and P retention and P CR increased linearly with phytase addition.

### 3.3. Bone Mineralization and Blood Analyses

The effect of dietary treatment on bone mineralization and blood mineral concentration traits is presented in Table 5. In the short-term trial, neither the mineral level nor the inclusion of phytase in the diet significantly affected any of the mineralization parameters that we analyzed in tibias at 38 d of age, after 17 days of receiving the experimental diets. However, when the animals received the experimental diets for 38 days (long-term trial), chicks fed with the PC diet showed a higher tibia weight and ash, Ca, and P content in tibia than those fed the NC diet (*p* < 0.05). Regarding phytase levels, animals fed with the NC diet showed a lower tibia weight and ash, Ca, and P content in tibia than those fed phytase-added diets (on av. −0.69 ± 0.15, −0.36 ± 0.10, −0.14 ± 0.05, and −0.08 ± 0.03 g, respectively; *p* < 0.05). These parameters showed a significant linear tendency to increase with phytase addition (*p* < 0.05). Dietary treatment did not significantly affect Ca and P content in the blood at 38 days of age in the long-term trial.

## 4. Discussion

### 4.1. Mineral Levels

The effects of reducing Ca and P levels on performance were clear in the starter period for the long-term trial. In the growing period of both the long and the short-term trials, animals had similar growth performance (similar FCR), independently of the mineral level. In the long-term trial, chicks fed the low Ca and P levels from day 1 to 21 of age showed lower ADG and ADFI compared with the animals fed the required Ca and P. These differences disappeared in the growing period, suggesting a possible age effect. However, the duration of the trial did not affect the animals’ response to mineral-deficient diets in terms of performance since no differences were observed in ADG, ADFI, or FCR, neither in the short nor in the long-term trial, when the whole experimental period was considered. Some works in the literature also suggest that age has an influence on the utilization of nutrients by broilers, since the rate of nutrient utilization is greater in old compared with young animals [19,20,24,31]. This, and the fact that young animals are characterized by the rapid growth of organs and tissues, could explain that the consequences of mineral deficiency are more harmful in young compared with more mature animals. On the other hand, it must be taken into account that the sample size for growth performance parameters in the short-term trial was limited (*n* = 8) and a greater number of animals would perhaps be required to extract more solid conclusions. 

In terms of nutrient digestibility and retention, as expected, Ca and P excretion was greater in the PC compared with the NC treatment in both trials. However, Ca and P CR were greater in the NC animals compared with the PC, although only significant in the case of P in the long-term trial. This finding could indicate that chicks fed mineral-deficient diets try to compensate for their deficiency by increasing retention, and this effect seems to be more important as the duration of the deficiency is longer. In fact, Ca and P CR values were both higher at the same age in the long than in the short-term trial. In this regard, other studies in broilers also suggest that feeding low-P diets during long feeding periods might lead to homeostatic adaptations in the digestive ability of the birds that increase P retention [19,23]. Despite the increase in P digestibility with the NC diet in the long-term trial, this mechanism was not enough to sustain the same level of tibia mineralization compared with animals fed the PC diet. In the short-term trial, bone mineralization was not significantly affected by Ca and P deficiency. Many researchers reported that a reduction in dietary P could be achieved without deleterious effects on bone mineralization if Ca is reduced concomitantly, and the reason might be that the Ca to P ratio is still within the range between 2:1 and 1:1, which is generally acceptable for the poultry industry [3]. In the present study, both Ca and P were concomitantly reduced, and the ratio was always greater than 1:1. This could be the reason for the lack of differences in tibia mineralization in the short-term trial. Therefore, the duration of the trial might increase the impact of feeding low-mineral diets (independently of the final Ca to P ratio), with longer periods being more damaging for bone mineralization in broilers than shorter ones. 

The Ca and P content in serum and bone can well reflect the nutritional status of Ca and P in the bodies of broilers. Some studies reported a decrease in the Ca and P serum levels when the mineral content in diets was lower [22,32]. However, in the present study, serum Ca and P concentrations determined in the long-term trial were numerically, although not significantly, greater in animals fed the NC diet. This agrees with the greater digestibility of these minerals, particularly P, in the NC compared with the PC of the long-term trial. 

### 4.2. Phytase Level 

As for the mineral content of the diets, growth performance was affected by the phytase concentration in feeds only during the starter phase of the long-term trial. In this regard, the group of animals receiving 500 and 1000 FTU/kg feed of the new phytase showed lower ADFI and FCR compared with NC from 1 to 21 days of age. As birds’ feed intake is based on the dietary energy level, the higher availability of some nutrients (including energy) in diets with added phytase could explain the reduced intake and improved conversion rate of animals fed with added phytases. Moreover, an age effect on phytase efficacy, according to which this effect is more evident in young animals due to the immaturity of the digestive enzymes and acids, has been reported by other studies in broilers [19,20,33] and in pigs [29]. 

On the other hand, the effect of phytase inclusion on nutrient digestion was particularly evident in the short-term trial compared with the long-term trial. In the case of mineral digestion, phytase addition improved the CR of Ca and P already at the lowest phytase dosage (250 FTU/kg) in the short-term trial, compared with the NC treatment. In the long-term trial, this improvement was observed at 500 FTU/kg, and only for P. Additionally, the inclusion of 250 and 500 FTU/kg feed improved the DM, OM, CP, and GE CR compared with the NC group in the short-term trial, but this improvement was not observed in the long-term trial. In this case, as the nutrient balance was carried out in animals of the same age in both trials, the difference can be attributed to the duration of the trial. As has been mentioned for the mineral level, feeding low-P diets during long feeding periods might lead to physiological adaptations in the digestive ability of the birds that increase P retention [19,22,23,24,31]. Part of this adaptation is related to improved phytate P digestion [23]. In fact, the mineral digestibility of the NC group was greater in the long-term compared with the short-term trial, decreasing the room for improvement of the new phytase versus the NC group. This effect has important implications for the final effective doses determined for phytases. In the present study, the final effective doses for the ePhyt 1000^®^ 3-phytase tested to increase P digestibility was 250 FTU/kg in the short-term trial and 500 FTU/kg in the long-term trial. The duration of the trial should therefore be considered when designing trials for evaluating the efficacy of a given phytase.

In terms of bone mineralization, as expected, the effect of phytase addition was more marked in the long-term trial. In this trial, animals fed NC with phytase showed increased tibia weight and ash, Ca, and P content (g) in tibia compared with NC at the lowest phytase doses (250 FTU/kg). In fact, animals fed with phytase were able to recover the levels of the PC for these parameters, contrary to animals from the short-term trial. Therefore, it seems that a longer period of phytase administration is needed to observe responses in tibia mineralization traits. Mineral content in serum was not different among treatments, although, in the case of P, it grew numerically with the inclusion of phytase. In this regard, other studies also observed increases in blood P concentration with phytase inclusion as a result of its effects on releasing P from the phytate [19,34].

Thus, the evaluation of parameters such as growth performance, nutrient utilization, and bone mineralization has been proven valuable in determining mineral deficiency consequences and the efficiency of phytase in improving P bioavailability, as suggested by previous studies [9,10,12,35,36]. However, the effective doses of the new phytase assessed in this study can vary from 250 to 500 FTU/kg, depending on the design of the trial and the response criteria selected.

## 5. Conclusions

The results of this work permit us to conclude that the evaluation of parameters such as growth performance, nutrient utilization, and bone mineralization has been proven valuable in determining mineral deficiency consequences and the efficiency of phytase in improving P bioavailability. However, the age and duration of the trial can differently affect these response criteria. Indeed, the effects of providing mineral (Ca and P)-deficient phytase-supplemented diets on growth performance depend on the age of the animals, being more noticeable in young animals. In terms of mineral and nutrient digestibility, the duration of the trial is key due to the adaptation phenomena of birds to low P supplies by increasing its digestibility and retention. Bone mineralization recovery seems also to depend on the duration of the trial, with longer periods being more effective for the bone accumulation of minerals. The result from this study demonstrates the necessity of standardization of these factors in future studies using minerals and phytase.

## Figures and Tables

**Table 1 animals-12-01418-t001:** Ingredients and analyzed chemical composition of positive control (PC) and negative control (NC) diets (% as-fed basis) in short- and long-term trials.

	Short-Term Trial		Long-Term Trial			
	Grower Feed		Starter Feed		Grower Feed	
	PC	NC	PC	NC	PC	NC
Ingredients %						
Corn grain	25.6	25.7	19.1	19.3	25.6	25.9
Wheat grain	34.6	35.0	34.6	35	34.6	35.0
Soybean meal 44% CP	15	15.1	21.2	21.3	15	15.1
Extruded soybean meal	17.9	18	14.9	15	17.9	18
Soybean oil	2.81	2.81	5.6	5.6	2.81	2.81
L-lysine	0.41	0.41	0.5	0.5	0.41	0.41
DL-methionine	0.28	0.28	0.35	0.35	0.28	0.28
L-threonine	0.10	0.10	0.13	0.13	0.10	0.10
Calcium carbonate	0.88	0.78	0.73	0.63	0.74	0.64
Dicalcium phosphate	1.65	0.95	1.8	1.1	1.65	0.95
Salt	0.23	0.23	0.21	0.21	0.23	0.23
Sodium bicarbonate	0.22	0.22	0.3	0.3	0.22	0.22
Vitamin–mineral premix ^1^	0.4	0.4	0.6	0.6	0.4	0.4
Chemical composition %						
Dry matter	89.7	89.4	90.8	90.5	90.7	90.7
Ash	5.25	4.62	5.01	4.66	4.89	4.56
Crude protein	18.1	17.8	22.6	23.1	22.3	22.3
Ether extract	3.47	4.10	3.98	3.67	3.28	4.16
Gross energy (kcal/kg)	4119	4206	4150	4244	4020	4202
AME (kcal/kg) ^2^	2863	2917	2884	2943	2794	2914
Calcium	0.91	0.68	0.97	0.73	0.98	0.70
Phosphorous	0.60	0.50	0.65	0.58	0.66	0.52
Phytate–phosphorous ^3^	0.21	0.18	0.24	0.23	0.23	0.24

^1^ Provides per kilogram of premix: calcium: 200.61 g, E5 manganese (manganese oxide): 13,000 mg, E6 zinc (zinc oxide): 7400 mg, E4 copper (copper sulphate pentahydrate): 800 mg, E2 iodine (potassium iodide): 380 mg, E8 selenium (sodium selenite): 20 mg, E1 iron (carbonate ferrous): 3600 mg, E672 vitamin A: 1,500,000 UI, E671 vitamin D3: 300,000 UI, vitamin K: 300 mg, vitamin B2: 600 mg, vitamin B12: 2000 mg, niacin: 3000 mg, calcium pantothenate: 1400 mg, pantothenic acid: 1288 mg, betaine: 10,830 mg, choline chloride: 25,500 mg, E320 butylhydroxyanisol (BHA): 4 mg, E321 butylhidroxytoluene (BHT): 44 mg, E324 ethoxyquin: 6.40 mg, dry matter: 956.54 g. ^2^ Apparent metabolizable energy: calculated from the average coefficients of total tract apparent digestibility of gross energy obtained for the PC and NC in this study. ^3^ Calculated as the difference between total phosphorus and phytate phosphorus.

**Table 2 animals-12-01418-t002:** Short-term trial: Initial and final body weight (BW, g), daily weight gain (ADG, g/d), daily feed intake (ADFI, g/d), and feed conversion ratio (FCR, g feed/g weight) of broilers fed feeds including different levels of phytase from 21 to 38 days of age.

	Dietary Treatment						*p*-Value	
	PC	NC	P250	P500	P1000	SEM	Treatment	Linear
BW, 21 d	897	867	853	861	855	16	0.375	0.191
BW, 38 d	2477	2486	2421	2419	2530	59	0.455	0.912
ADG, 28 to 38 d	98.0	99.1	94.9	95.4	99.0	4.6	0.940	0.768
ADFI, 28 to 38 d	163	162	155	153	161	7.0	0.766	0.977
FCR, 28 to 38 d	1.67	1.64	1.64	1.62	1.63	0.03	0.793	0.390

Treatments: PC, positive control; NC, negative control; P250, negative control with phytase at 250 FTU/kg feed; P500, negative control with phytase at 500 FTU/kg feed; and P1000, negative control with phytase at 1000 FTU/kg feed. Data represent mean values of 8 replicate cages of two chicks each per treatment. SEM: Standard error of the mean.

**Table 3 animals-12-01418-t003:** Long-term trial: Initial and final body weight (BW, g), daily weight gain (ADG; g/d), daily feed intake (ADFI, g/d), and feed conversion ratio (FCR, g feed/g weight) of broilers fed feeds including different levels of phytase from 1 to 42 days of age.

	Dietary Treatment						*p*-Value	
	PC	NC	P250	P500	P1000	SEM	Treatment	Linear
BW, 1 d	45.2	44.8	44.6	45.6	45.4	0.327	0.161	0.087
BW, 42 d	3297	3366	3324	3329	3328	41	0.808	0.648
Starter period, 1 to 21 d:								
ADG	48.6 ^b^	45.6 ^a^	43.4 ^a^	45.8 ^a^	46.9 ^ab^	0.734	<0.001	0.035
ADFI	65.3 ^c^	62.1 ^b^	61.3 ^b^	56.3 ^a^	57.0 ^a^	0.794	<0.001	<0.001
FCR	1.34 ^b^	1.36 ^b^	1.41 ^c^	1.23 ^a^	1.22 ^a^	0.013	<0.001	<0.001
Growing period, 22 to 42 d:								
ADG	105.7	110.2	110.9	109.1	107.4	1.8	0.252	0.185
ADFI	168.8	174.3	172.2	166.9	172.1	2.86	0.388	0.880
FCR	1.60	1.58	1.55	1.53	1.60	0.024	0.158	0.197
Global period, 1 to 42 d:								
ADG	75.2	75.5	74.6	75.2	74.8	0.99	0.965	0.777
ADFI	113.7 ^b^	114.0 ^b^	112.6 ^ab^	107.8 ^a^	110.3 ^ab^	1.6	0.051	0.130
FCR	1.51 ^b^	1.51 ^b^	1.51 ^b^	1.43 ^a^	1.47 ^ab^	0.02	0.013	0.116

^a, b, c^ Least square means in a row not sharing superscripts differ at *p* < 0.05. Treatments: PC, positive control; NC, negative control; P250, negative control with phytase at 250 FTU/kg feed; P500, negative control with phytase at 500 FTU/kg feed; and P1000, negative control with phytase at 1000 FTU/kg feed. Data represent mean values of 7 replicate pens per treatment. SEM: Standard error of the mean.

**Table 4 animals-12-01418-t004:** Effect of mineral and phytase inclusion levels on nutrient coefficient of retention (CR, %), retention (g/d animal), and excretion (g/d animal) of broilers in both short- and long-term trials (35 to 38 days of age).

	Dietary Treatment						*p*-Value	
	PC	NC	P250	P500	P1000	SEM	Treatment	Linear
Short-term trial:								
DM CR	67.4 ^ab^	66.2 ^a^	68.8 ^b^	68.7 ^b^	68.2 ^b^	0.4	<0.001	0.011
OM CR	70.0 ^ab^	68.4 ^a^	70.3 ^b^	70.9 ^b^	70.1 ^b^	0.4	0.006	0.026
CP CR	58.9 ^ab^	56.0 ^a^	61.0 ^b^	61.0 ^b^	58.3 ^ab^	0.9	0.001	0.312
GE CR	70.2 ^ab^	69.3 ^a^	70.6 ^ab^	71.5 ^b^	70.9 ^ab^	0.6	0.146	0.077
Ca CR	28.8 ^a^	33.6 ^a^	41.0 ^b^	43.0 ^b^	40.7 ^b^	1.9	<0.001	0.014
Ca retention	0.46 ^ab^	0.40 ^a^	0.52 ^bc^	0.55 ^c^	0.54 ^bc^	0.03	0.004	0.004
Ca excretion	1.16 ^b^	0.71 ^a^	0.75 ^a^	0.73 ^a^	0.79 ^a^	0.05	<0.001	0.748
P CR	34.2 ^a^	38.8 ^a^	46.3 ^b^	46.6 ^b^	45.3 ^ab^	2.0	<0.001	0.058
P retention	0.35 ^a^	0.33 ^a^	0.41 ^ab^	0.42 ^b^	0.42 ^b^	0.02	0.019	0.023
P excretion	0.72 ^b^	0.51 ^a^	0.47 ^a^	0.48 ^a^	0.51 ^a^	0.04	<0.001	0.920
Long-term trial:								
DM CR	68.4	67.9	69.3	68.2	68.0	0.6	0.435	0.656
OM CR	70.1	68.8	70.6	69.7	69.5	0.6	0.278	0.800
CP CR	60.4	59.0	59.7	57.9	58.1	0.9	0.193	0.266
GE CR	68.8	69.4	70.4	68.0	69.1	0.7	0.258	0.443
Ca CR	31.1 ^a^	36.0 ^ab^	34.6 ^ab^	41.8 ^b^	39.4 ^b^	2.0	0.005	0.090
Ca retention	0.59 ^b^	0.47 ^a^	0.50 ^a^	0.68 ^c^	0.63 ^bc^	0.03	<0.001	<0.001
Ca excretion	1.31 ^b^	0.85 ^a^	0.94 ^a^	0.96 ^a^	0.96 ^a^	0.05	<0.001	0.149
P CR	36.8 ^a^	42.1 ^b^	40.2 ^ab^	46.9 ^c^	45.8 ^bc^	1.4	<0.001	0.010
P retention	0.46 ^ab^	0.43 ^a^	0.40 ^a^	0.50 ^b^	0.48 ^ab^	0.02	0.009	0.009
P excretion	0.81 ^b^	0.56 ^a^	0.59 ^a^	0.57 ^a^	0.60 ^a^	0.03	<0.001	0.387

^a, b, c^ Least square means in a row not sharing superscripts differ at *p* < 0.05. Treatments: PC, positive control; NC, negative control; P250, negative control with phytase at 250 FTU/kg feed; P500, negative control with phytase at 500 FTU/kg feed; and P1000, negative control with phytase at 1000 FTU/kg feed. Data represent mean values of 8 replicate cages of two chicks each per treatment. SEM: Standard error of the mean.

**Table 5 animals-12-01418-t005:** Effect of mineral level and dietary phytase inclusion on bone mineralization traits and blood concentration of calcium (Ca) and phosphorus (P) of broilers in both short- (at 38 days of age after 17 days of treatment) and long-term trials (at 38 days of age after 38 days of treatment).

	Dietary Treatment						*p*-Value	
	PC	NC	P250	P500	P1000	SEM	Treatment	Linear
Short-term trial:								
Tibia weight, g	5.27	4.72	4.74	4.57	4.91	0.26	0.348	0.591
Tibia weight, % body weight	0.22	0.19	0.20	0.18	0.19	0.01	0.330	0.886
Ash in tibia, g	2.80	2.44	2.49	2.38	2.60	0.13	0.193	0.434
Ash in tibia, % dry matter	53.1	51.8	52.5	52.2	52.8	0.4	0.155	0.107
Ca in tibia, g	1.05	0.91	0.92	0.89	0.98	0.05	0.157	0.344
Ca in tibia, % dry matter	19.0	19.4	19.5	19.4	19.7	0.2	0.187	0.223
P in tibia, g	0.49	0.44	0.44	0.41	0.44	0.03	0.386	0.891
P in tibia, % dry matter	9.24	9.28	9.18	9.07	9.00	0.134	0.514	0.113
Long-term trial:								
Tibia weight, g	4.82 ^b^	4.21 ^a^	4.85 ^b^	4.93 ^b^	4.91 ^b^	0.14	0.005	0.005
Tibia weight, % body weight	0.18 ^ab^	0.17 ^a^	0.18 ^b^	0.19 ^b^	0.18 ^ab^	0.01	0.061	0.138
Ash in tibia, g	2.60 ^b^	2.23 ^a^	2.61 ^b^	2.55 ^b^	2.62 ^b^	0.08	0.005	0.007
Ash in tibia, % dry matter	53.9	53.0	53.9	53.2	53.3	0.4	0.295	0.902
Ca in tibia, g	0.99 ^b^	0.86 ^a^	1.00 ^b^	1.01 ^b^	1.00 ^b^	0.04	0.016	0.016
Ca in tibia, % dry matter	20.7	20.3	20.5	20.3	20.4	0.181	0.385	0.835
P in tibia, g	0.47 ^b^	0.40 ^a^	0.47 ^b^	0.48 ^b^	0.48 ^b^	0.02	0.008	0.009
P in tibia, % dry matter	9.71	9.54	9.73	9.70	9.67	0.09	0.512	0.424
Minerals in blood:								
Ca, mg/dL	11.9	12.1	11.8	12.5	11.7	0.4	0.510	0.575
P, mg/dL	9.2	10.4	9.3	9.8	10.2	0.4	0.173	0.876

^a, b, c^ Least square means in a row not sharing superscripts differ at *p* < 0.05. Treatments: PC, positive control; NC, negative control; P250, negative control with phytase at 250 FTU/kg feed; P500, negative control with phytase at 500 FTU/kg feed; and P1000, negative control with phytase at 1000 FTU/kg feed. Data represent mean values of 8 replicate animals per treatment. SEM: Standard error of the mean.

## Data Availability

Data are contained within the article.

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
