# Peer review of "The Duration of the Trial Influences the Effects of Mineral Deficiency and the Effective Phytase Dose in Broilers’ Diets"

_animals, 2022, doi:10.3390/ani12111418_

Round 1
Reviewer 1 Report
Dear editor
Pls see the attached file.

Author Response
Response to the comments of Reviewer 1:
The authors thank the reviewer for the comments made on the manuscript. We have considered all the comments in order to improve the quality of the paper, and a point-by-point response to your comments are provided in the following lines (responses in red):
- This is not originally study.
The novelty of the study relies on the fact that its main objective is to study how traditional criteria response to phytase addition in diets (growth performance, bone mineralization, digestibility,…) are influenced by a less studied factor such as the duration of the trial. The results obtained led us to suggest the necessity of standardization of these factors in future studies using minerals and phytase.
- It is wrong to cite the source in the text. There are many mistakes.
Batabunde et al. (2021) [13] WRONG
Batabunde et al. [13] TRUE
Shirley and Edwards (2003) [17] WRONG
Shirley and Edwards [17] TRUE
The mistakes on the citations have been checked and solved along the manuscript.
- INTRODUCTION does not make enough references to the latest literature on the subject. Old references should be taken out.
Some old references have been replaced by more recent ones (reference 5 of the original version) or taken out of the manuscript (reference 2 of the original version). Other more recent references have been added in the introduction, materials and methods and discussion sections such as Abd El-Hack et al. (2018), Dersjant-Li et al. (2018), Alam et al. (2020), Batabunde et al. (2022) and Zhang et al. (2022).
- Much of what is described in the introduction is a well-known topic.
The novelty of the study relies on the investigation of a less studied factor (duration of the trial) which can seriously affect the optimum phytase level in diets.
- The experimental methods are appropriate for this study and stated clearly.
ok
- At the bottom of the table 1, what is expressed by 1 and 2 should be written.
Vitamin-mineral premix1
AME (kcal/kg)2
Information referred to superscripts 1 and 2 were added in Table 1.
- Either use ‘feed intake’ or ‘feed consumption uniformly throughout the text.
“Feed consumption” has been replaced by “feed intake” in L162 of the revised manuscript
- 2.5. Bone and Blood Sampling: GIVE RELEVANT REFERENCE
L187 of the revised manuscript: the reference Dersjant-Li et al. (2018) has been included.
- 2.4. Total Tract Digestibility; GIVE RELEVANT REFERENCE
L177 of the revised manuscript: the reference Dersjant-Li et al. (2018) has been included.
- 2.7. Statistical Analyses
The CTTAD of DM, organic matter (OM), GE, CP, Ca and P were calculated using the following equation: ????? (%)= [(???? ?????? × ????????????)−(??????? ?????? × ???????????????)](???? ?????? × ????????????)×100 GIVE RELEVANT REFERENCE
L211 of the revised manuscript: the reference Javadi et al. (2021) has been added here.
- Results are enough.
ok
- More discussion is required.
In the work, the main novel results obtained have been discussed. In any case, the entire discussion has been reviewed, trying to see if anything else could be added to improve the discussion. In this sense, an attempt has been made to improve the discussion corresponding to the effect of the level of minerals.
- Uniformity in references must be seen. Initial letters must be lowercase
Example:
Javadi, M.; Pascual, J.J.; Cambra-López, M.; Macías-Vidal, J.; Donadeu, A.; Dupuy, J.; Carpintero, L.; Ferrer, P.; Cerisuelo, A. 446 Effect of Dietary Mineral Content and Phytase Dose on Nutrient Utilization, Performance, Egg Traits and Bone Mineralization 447 in Laying Hens from 22 to 31 Weeks of Age. Animals 2021, 11, 1495.
References have been homogenized in the references section.
- References are not enough.
Six new references have been added in the text.

Reviewer 2 Report
Thank you for giving me an opportunity to review the paper "The Duration of the Trial Influences the Effects of Mineral Deficiency and the Effective Phytase Dose in Broilers’ Diets". The paper is interesting and provides some novel aspects which were not covered previously. The results are important for poultry producers. The title of the paper is well written, abstract is of sufficient length, introduction is indepth. Matrials and methods are comprehensive, results are concize and discussion is of sufficent length. In my opinion, the paper is acceptable after incorporating the following basic studies.
The uses of microbial phytase as a feed additive in poultry nutrition – a review. Annals of Animal Science 18: 639-658.
Effect of Bacillus cereus and phytase on the expression of musculoskeletal strength and gut health in Japanese quail (Coturnix japonica). Journal of Poultry Science, 57:200-204.
Author Response
Thank you for giving me an opportunity to review the paper "The Duration of the Trial Influences the Effects of Mineral Deficiency and the Effective Phytase Dose in Broilers’ Diets". The paper is interesting and provides some novel aspects which were not covered previously. The results are important for poultry producers. The title of the paper is well written, abstract is of sufficient length, introduction is indepth. Matrials and methods are comprehensive, results are concize and discussion is of sufficent length. In my opinion, the paper is acceptable after incorporating the following basic studies.
The uses of microbial phytase as a feed additive in poultry nutrition – a review. Annals of Animal Science 18: 639-658.
Effect of Bacillus cereus and phytase on the expression of musculoskeletal strength and gut health in Japanese quail (Coturnix japonica). Journal of Poultry Science, 57:200-204.
The authors thank the comments made by the reviewer and have included the two studies suggested by the reviewer in the text of the manuscript and in the “References” section. L 63 and L 71 of the revised manuscript (in red).

Reviewer 3 Report
Manuscript animals-1708192, entitled “The Duration of the Trial Influences the Effects of Mineral Deficiency and the Effective Phytase Dose in Broilers’ Diets”
Recommendation: The above paper is not suitable for publication in its present form.
The article provides useful information about the effects of mineral deficiency and the effective phytase dose in broilers’ diets according to the experimental duration. Although, the experiment was in general appropriately designed and implemented, there are some points that should be corrected or clarified.
General comments
- Please provide the experimental unit. In general, it is not the animal but the metabolic cage (L204-206, 299).
- Please explain in section 2.3 that ADG, ADFI and FCR were calculated after 28 day in short-term trial.
- Why was a finisher diet not used?
- How was the mortality rates calculated in long-term trial? Number of birds was 490 (L100) or 80 (L106)? Mortality rates are presented after the age of 28 days?
- In several parts (L33-34, 254, 260-261, 275-276, a linear effect is presented. However, P-linear is not provided. Please add it to the Tables and Text.
- L287-289: Why do you consider the effects as negative? FCR remained similar.
- Please delete year (in parentheses) after the name of the researcher (L63, 67, 73, 131, 136, 178, 182)
Minor points
L25: “varying” instead of “differing”
L26: “sufficient” instead of “correct”
L30: Please be specific. What do you mean by “lower performance”?
L37: “supplementary levels” instead of “supplies”
L42: “necessary” instead of “required”
L43: “…participates in metabolic…”
L48: “reservoir” instead of “reserve”
L49: “Due to the” instead of “Because of”
L50: “As a result” instead of “Then”
L52: “…or include enzymes, such as phytases.”
L56: “As a consequence, an increase in weight gain and bone ash percentages in broilers is observed.”
L57-58: “…to 12000 FTU/kg, with the inclusion level around 500 FTU/kg being the most common…”
L58-59: “The effective dose considered as the minimum at which positive effects are observed is generally adjusted according…”
L62: “However, the interaction of phytases with these parameters…”
L63-64: “…found a quadratic effect of phytase on BW…”
L71-72: “…that prolonging the duration of phytase supplementation using P deficient…”
L82: Please add a reference
L86-87: “…phytase doses by comparing a short…”
L90: “varying” instead of “differing”
L91-92: “…of 21 days of age with a mean body weight (BW) of 872 ± 27.4 g (standard deviation; SD) were used.”
L93: “allocated into” instead of “distributed in”
L97: “removed” instead of “housed”
L100-101: “of age with a mean BW of 45.0 ± 0.93 (SD) g were used.”
L102: “allotted to” instead of “distributed in”
L107: “…to their weight…”
L120: “The applied light regime” instead of “Light program provided”
L124, 127, 130, 133, 136, 163: “diets” instead of “feeds”
L134: “…three diets in which…”
L135-136: “…feed, respectively. Details regarding the used phytase are provided in the study by Salaet et al. [23].”
L149: “intake” instead of “consumption”
L164: “collected” instead of “taken”
L173: “determine” instead of “analyze”
L184: “…described by Haugh and Lantzch [26].”
L186-187: “Afterwards, tibias were left in an ether…”
L210: What do you mean by “the initial pen weight”?
L215: “adequate” instead of “good”
L233: “during the whole experimental period” instead of “globally”
L238: “whole” instead of “global”
L247-248: “…but similar CTTAD of nutrients and Ca and P retention than…”
L266: Please delete “Finally”
L282-285: Please delete. It is not necessary
L292-293: What do you mean? Please explain
L299: “…trial was limited...” I think that n=8
L304: “This finding could indicate…”
L334-335: “As birds feed intake is based on…”
L343-344: Only at the lowest dose?
L345: “observed” instead of “seen starting”
L347-348: Please delete “in other nutrients digestibility”
L372, 379: “…have been proved…”
L381-382: “…can differently affect these response…”
L382-383: “…deficient phytase supplemented diets on growth…”
L387: “…of the trial, with longer periods being more effective…”
Author Response
Response to the comments of Reviewer 3:
The authors thank the reviewer for the comments made on the manuscript. We have considered all the comments in order to improve the quality of the paper, and a point-by-point response to your comments is provided in the following lines (responses in red):
Manuscript animals-1708192, entitled “The Duration of the Trial Influences the Effects of Mineral Deficiency and the Effective Phytase Dose in Broilers’ Diets”
Recommendation: The above paper is not suitable for publication in its present form.
The article provides useful information about the effects of mineral deficiency and the effective phytase dose in broilers’ diets according to the experimental duration. Although, the experiment was in general appropriately designed and implemented, there are some points that should be corrected or clarified.
General comments
- Please provide the experimental unit. In general, it is not the animal but the metabolic cage (L204-206, 299).
The experimental units considered for each parameter (pen, cage and animal) are mentioned in L222-224 and L318 of the revised manuscript.
- Please explain in section 2.3 that ADG, ADFI and FCR were calculated after 28 day in short-term trial.
L164-166. In section 2.3 the period used to calculate growth performance in each trial was defined.
- Why was a finisher diet not used?
We decided to implement a two-phase feeding program to simplify feed formulation and the comparison between the short-term and the long-term trial.
- How was the mortality rates calculated in long-term trial? Number of birds was 490 (L100) or 80 (L106)? Mortality rates are presented after the age of 28 days?
Mortality rates were calculated as (number of animals dead/total initial number of animals)*100. The total number of birds in the long-term trial was 490, but 80 of them were used for determining nutrient balance. Mortality rates were calculated from day 21 of age in the short-term trial and from day 1 of age in the long-term trial.
- In several parts (L33-34, 254, 260-261, 275-276, a linear effect is presented. However, P-linear is not provided. Please add it to the Tables and Text.
The P-value of the linear contrast has been added to tables 2, 3, 4 y 5 and in the text (L255, L273, L280-281 of the revised manuscript).
- L287-289: Why do you consider the effects as negative? FCR remained similar.
L304-306 of the revised manuscript. This sentence has been rephrased in order is transmit that animals had a similar growth performance (similar FCR).
- Please delete year (in parentheses) after the name of the researcher (L63, 67, 73, 131, 136, 178, 182)
The year has been deleted after the name of the authors (L 64, 68, 74, 132, 137, 199 and 201)
Minor points
L25: “varying” instead of “differing”
L25 of the revised manuscript: “Differing” has been replaced by “varying”
L26: “sufficient” instead of “correct”
L26 of the revised manuscript: “Correct” has been replaced by “sufficient”
L30: Please be specific. What do you mean by “lower performance”?
L30-31 of the revised manuscript: “Lower performance” has been replaced by “chicks fed NC diets showed a lower (p<0.05) average daily gain and feed intake compared to chicks fed PC and a greater (p<0.05) feed conversion ratio compared to 500 and 1000 FTU/kg feed during the starting period”.
L37: “supplementary levels” instead of “supplies”
L38 of the revised manuscript: “Supplies” has been replaced by “supplementary levels”.
L42: “necessary” instead of “required”
L42 of the revised manuscript: “Required” has been replaced by “necessary”
L43: “…participates in metabolic…”
L44 of the revised manuscript: “of” has been replaced by “in”
L48: “reservoir” instead of “reserve”
L49 of the revised manuscript: “reserve” has been replaced by “reservoir”
L49: “Due to the” instead of “Because of”
L50 of the revised manuscript: “Because of” has been replaced by “Due to the”
L50: “As a result” instead of “Then”
L51 of the revised manuscript: “Then” has been replaced by “As a result”
L52: “…or include enzymes, such as phytases.”
L53 of the revised manuscript: “include phytases” has been replaced by “include enzymes, such as phytases”
L56: “As a consequence, an increase in weight gain and bone ash percentages in broilers is observed.”
L57-58 of the revised manuscript: “This derives in increases in weight gain and bone ash percentages in broilers” has been replaced by “As a consequence, an increase in weight gain and bone ash percentages in broilers is observed”
L57-58: “…to 12000 FTU/kg, with the inclusion level around 500 FTU/kg being the most common…”
L58-59 of the revised manuscript: “to 12000 FTU/kg, being an inclusion level around 500 FTU/kg the most common” has been replaced by “to 12000 FTU/kg, with the inclusion level around 500 FTU/kg being the most common”
L58-59: “The effective dose considered as the minimum at which positive effects are observed is generally adjusted according…”
L59-61 of the revised manuscript: “This effective dose (minimum dose at which positive effects are observed) is generally fixed according to” has been replaced by “The effective dose considered as the minimum at which positive effects are observed is generally adjusted according”
L62: “However, the interaction of phytases with these parameters…”
L63-64 of the revised manuscript: “However, the response of phytases in these parameters” has been replaced by “However, the interaction of phytases with these parameters…”
L63-64: “…found a quadratic effect of phytase on BW…”
L64-65 of the revised manuscript: “found a quadratic response to phytase in BW” has been replaced by “…found a quadratic effect of phytase on BW…”
L71-72: “…that prolonging the duration of phytase supplementation using P deficient…”
L72-73 of the revised manuscript: “that prolonging phytase supplementation duration using P deficient” has been replaced by “that prolonging the duration of phytase supplementation using P deficient”
L82: Please add a reference
L83 of the revised manuscript: a reference has been added at this point.
L86-87: “…phytase doses by comparing a short…”
L87-88 of the revised manuscript: “phytase doses comparing a short” has been replaced by “phytase doses by comparing a short”
L90: “varying” instead of “differing”
L91 of the revised manuscript: “differing” has been replaced by “varying”
L91-92: “…of 21 days of age with a mean body weight (BW) of 872 ± 27.4 g (standard deviation; SD) were used.”
L92-93 of the revised manuscript: “of 21 days of age and 872 ± 27.4 (standard deviation; SD) g body weight (BW) were used” has been replaced by “of 21 days of age with a mean body weight (BW) of 872 ± 27.4 g (standard deviation; SD) were used”
L93: “allocated into” instead of “distributed in”
L94 of the revised manuscript: “distributed in” has been replaced by “allocated”
L97: “removed” instead of “housed”
L98 of the revised manuscript: “housed” has been replaced by “removed”
L100-101: “of age with a mean BW of 45.0 ± 0.93 (SD) g were used.”
L101-102 of the revised manuscript: “of age and 45.0 ± 0.93 (SD) g BW were used” has been replaced by “of age with a mean BW of 45.0 ± 0.93 (SD) g were used”
L102: “allotted to” instead of “distributed in”
L103 of the revised manuscript: “distributed in” has been replaced by “allotted to”
L107: “…to their weight…”
L108 of the revised manuscript: “to its weight” has been replaced by “to their weight”
L120: “The applied light regime” instead of “Light program provided”
L121 of the revised manuscript: “Light program provided” has been replaced by “The applied light regime”
L124, 127, 130, 133, 136, 163: “diets” instead of “feeds”
L125, 128, 131,134, 137, 180 of the revised manuscript: “feeds” has been replaced by “diets”
L134: “…three diets in which…”
L135 of the revised manuscript: “three diets for which” has been replaced by “three diets in which”
L135-136: “…feed, respectively. Details regarding the used phytase are provided in the study by Salaet et al. [23].”
L136-137 of the revised manuscript: “feed, respectively (see enzyme details at Salaet et al. [23]).” has been replaced by “feed, respectively. Details regarding the used phytase are provided in the study by Salaet et al. [23].”
L149: “intake” instead of “consumption”
L162 of the revised manuscript: “consumption” has been replaced by “intake”
L164: “collected” instead of “taken”
L180 of the revised manuscript: “taken” has been replaced by “collected”
L173: “determine” instead of “analyze”
L191 of the revised manuscript: “analyze” has been replaced by “determine”
L184: “…described by Haugh and Lantzch [26].”
L201 of the revised manuscript: “described in Haugh and Lantzch [26].” has been replaced by “described by Haugh and Lantzch [26].”
L186-187: “Afterwards, tibias were left in an ether…”
L203-204 of the revised manuscript: “After, tibias were degreased in an ether” has been replaced by “Afterwards, tibias were left in an ether”
L210: What do you mean by “the initial pen weight”?
L222-224 of the revised manuscript: “the initial pen weight” is the weight of all animals of each pen on day 1 of the experimental period. This was used to covariate the rest of the weights in order to correct for initial variability. “The initial pen weight” was replaced by “the initial weight of the animals of each pen” in order to clarify.
L215: “adequate” instead of “good”
L233 of the revised manuscript: “good” has been replaced by “adequate”
L233: “during the whole experimental period” instead of “globally”
L252-253 of the revised manuscript: “globally” has been replaced by “during the whole experimental period”
L238: “whole” instead of “global”
L258 of the revised manuscript: “global” has been replaced by “whole”
L247-248: “…but similar CTTAD of nutrients and Ca and P retention than…”
L266-267 of the revised manuscript: “but similar CTTAD and retention than” has been replaced by “but similar CTTAD of nutrients and Ca and P retention than”
L266: Please delete “Finally”
L286 of the revised manuscript: “Finally” has been deleted.
L282-285: Please delete. It is not necessary
L301: The initial 4 lines of the discussion section were deleted.
L292-293: What do you mean? Please explain
L309-312 of the revised manuscript: Here, the sentence “However, the duration of the trial did not affect the animals’ response to mineral-deficient diets in terms of performance” means that performance parameters (ADG, ADFI and FCR) were similar between NC and PC group in both the short and long term trial, when the whole experimental period is considered. This sentence has been rephrased in order to clarify its meaning “However, the duration of the trial did not affect the animals’ response to mineral-deficient diets in terms of performance since no differences were observed in ADG, ADFI or FCR neither in the short nor in the long-term trial when the whole experimental period is considered”.
L299: “…trial was limited...” I think that n=8
L318 of the revised manuscript: “n=16” has been replaced by “n=8”
L304: “This finding could indicate…”
L323 of the revised manuscript: “This could indicate” has been replaced by “This finding could indicate”
L334-335: “As birds feed intake is based on…”
L352-353 of the revised manuscript: “As birds control their feed intake based on the dietary energy level” has been replaced by “As birds feed intake is based on”
L343-344: Only at the lowest dose?
L361-362 of the revised manuscript: with the sentence the authors refer to that mineral digestibility was improved already at the lowest phytase dose in the short-term trial. However, in the long-term trial a higher phytase dose is required to observe these effects. Then, “at the lowest phytase dosage” has been replaced by “already at the lowest phytase dosage” in order to clarify
L345: “observed” instead of “seen starting”
L363 of the revised manuscript: “seen starting” has been replaced by “observed”
L347-348: Please delete “in other nutrients digestibility”
“in other nutrients digestibility” has been deleted
L372, 379: “…have been proved…”
L389 and 396 of the revised manuscript: “have proved” has been replaced by “have been proved”
L381-382: “…can differently affect these response…”
L398-399 of the revised manuscript: “can affect differently to these response” has been replaced by “can differently affect these response”
L382-383: “…deficient phytase supplemented diets on growth…”
L399-400 of the revised manuscript: “deficient and including phytase diets on growth” has been replaced by “deficient phytase supplemented diets on growth”
L387: “…of the trial, with longer periods being more effective…”
L404 of the revised manuscript: “of the trial, being longer periods more effective” has been replaced by “of the trial, with longer periods being more effective”

Round 2
Reviewer 3 Report
Authors made all the necessary amendments and I suggest the acceptance of their article.
Please replace "Lineal" with "Linear" in Tables
Author Response
Response to the comments of Reviewer 3:
The authors thank the reviewer for the comments made on the manuscript. We have considered all the comments in order to improve the quality of the paper, and a point-by-point response to your comments is provided in the following lines. Additionally, all the revisions made to the manuscript have been marked up using the “Track Changes” function so that changes can be easily viewed by the editors and reviewers.
L4-14 of the revised manuscript. Names and affiliations have been carefully checked
Citation section. “The Duration of the Trial Influences the Effects of a Mineral Deficiency and the Effective Phytase Dose in Broilers’ Diets.” has been replaced by “The Duration of the Trial Influences the Effects of Mineral Deficiency and the Effective Phytase Dose in Broilers’ Diets.”
L244-246 of the revised manuscript. “feed intake, Nutrientfeed, excreta output, Nutrientexcreta” should be in italic to keep unified with equation
Table 2. Column 1 of Table 2 has been lightly modified (BW)
Table 3. Column 1 of Table 3 has been lightly modified (BW)
L598-599 of the revised manuscript. The reference FEDNA has been modified according to the version suggested by the reviewer
